# Macrophage Targeting pH Responsive Polymersomes for Glucocorticoid Therapy

**DOI:** 10.3390/pharmaceutics11110614

**Published:** 2019-11-15

**Authors:** Virgínia M. Gouveia, Loris Rizzello, Claudia Nunes, Alessandro Poma, Lorena Ruiz-Perez, António Oliveira, Salette Reis, Giuseppe Battaglia

**Affiliations:** 1LAQV/REQUIMTE, Department of Chemical Sciences, Faculty of Pharmacy, University of Porto, 4050-313 Porto, Portugal; virginia.mgouveia@gmail.com (V.M.G.); cdnunes@ff.up.pt (C.N.); 2Abel Salazar Biomedical Sciences Institute, University of Porto, Portugal, 4050-313 Porto, Portugal; afonsecaoliveira1@gmail.com; 3Department of Chemistry, University College London, 20 Gordon Street, London WC1H 0AJ, UK; lrizzello@ibecbarcelona.eu (L.R.); a.poma@ucl.ac.uk (A.P.); l.ruiz-perez@ucl.ac.uk (L.R.-P.); 4Institute of Physics of Living Systems, University College London, 20 Gordon Street, London WC1H 0AJ, UK; 5Institute for Bioengineering of Catalonia (IBEC), The Barcelona Institute of Science and Technology, Baldiri Reixac 10-12, 08028 Barcelona, Spain; 6Division of Biomaterials and Tissue Engineering, UCL Eastman Dental Institute, University College London, 256 Gray’s Inn Road, London WC1X 8LD, UK; 7EPSRC/JEOL Centre for Liquid Phase Electron Microscopy, University College London, 20 Gordon Street, London WC1H 0AJ, UK; 8Catalan Institution for Research and Advanced Studies (ICREA), Passeig Lluís Companys 23, 08010 Barcelona, Spain

**Keywords:** inflammation, macrophages, glucocorticoid, polymersomes

## Abstract

Glucocorticoid (GC) drugs are the cornerstone therapy used in the treatment of inflammatory diseases. Here, we report pH responsive poly(2-methacryloyloxyethyl phosphorylcholine)–poly(2-(diisopropylamino)ethyl methacrylate) (PMPC–PDPA) polymersomes as a suitable nanoscopic carrier to precisely and controllably deliver GCs within inflamed target cells. The in vitro cellular studies revealed that polymersomes ensure the stability, selectivity and bioavailability of the loaded drug within macrophages. At molecular level, we tested key inflammation-related markers, such as the nuclear factor-κB, tumour necrosis factor-α, interleukin-1β, and interleukin-6. With this, we demonstrated that pH responsive polymersomes are able to enhance the anti-inflammatory effect of loaded GC drug. Overall, we prove the potential of PMPC–PDPA polymersomes to efficiently promote the inflammation shutdown, while reducing the well-known therapeutic limitations in GC-based therapy.

## 1. Introduction

Inflammatory diseases comprise a vast range of disorders and conditions characterised by an uncontrolled over-activation of the immune system [1,2]. Inflammation is also indeed intimately related with the pathogenesis of many chronic autoimmune diseases, such as rheumatoid arthritis, lupus, psoriasis, and inflammatory bowel disorders [1,2]. Macrophages are a critical component of the innate immune system, and key cells in the initiation, maintenance, and resolution of inflammation [3,4,5]. These cells are professional phagocytes that are responsible for patrolling for pathogens, and modulating the inflammatory process [3,4,5]. Under inflammatory stimuli, macrophages become activated through the interaction of the Toll-like receptors (class of proteins that play a key role in the innate immune system) with specific cytokines, such as interleukin-1β (IL1β), interleukin-6 (IL6) and tumour necrosis factor-α (TNFα), and/or bacterial lipopolysaccharides [3,4,6]. Once activated, macrophages initiate the nuclear factor-κB (NF-κB) translocation signalling pathway, which is a common pathological mechanism in several inflammatory diseases [2,7]. Thereby, activated macrophages are tangled in inflammatory response through the production of a wide range of molecules, such as cytokines, chemokines, inflammatory enzymes, and growth factors [2,7]. These molecules, as an immune-defensive mechanism, can act as both pro- or anti-inflammatory mediators, and thus induce either deleterious effects or promote the resolution of inflammation [2,7]. Macrophages play then a critical role in the balance between pro- and anti-inflammatory mediators to ultimately shutdown the outcomes of the inflammatory response [1,2,7]. This role identifies macrophages as both local and systemic key players in inflammatory diseases, and henceforth their efficient targeting is a crucial aspect. Glucocorticoids (GCs) are the cornerstone therapy used in the management of several inflammatory diseases, because of their strong anti-inflammatory and immunosuppressive properties [7,8,9]. One of these drugs, prednisolone, is highly effective in the suppression of inflammation-activated signalling, such as the NF-κB pathway [7,8,9]. Consequently, the drug-induced NF-κB regulation leads to the suppression of multiple pro-inflammatory cytokines, such as IL1β, IL6, and TNFα, which foster the inflammatory response [7,8,10,11]. Despite the effective anti-inflammatory mechanism of action of GCs, the lack of selectivity towards the inflamed tissues bestows the use of high doses, which further induce deleterious side effects [8,9,12]. Additionally, some cases of drug-resistance have been reported in patients with severe asthma, as well as, in other chronic obstructive pulmonary diseases [8,12]. This inevitably compromises the anti-inflammatory efficacy of GC-based therapy. Despite the advances in nanomedicine proved to improve the delivery of GCs, the translation into the clinic remains a challenge [9]. There is still the need to overcome some drawbacks particularly related to: (*i*) the safety and physiologic stability of the carrier itself, (*ii*) the ability to effectively and controllably release loaded cargo and, henceforth, (*iii*) the intracellular drug dose effectiveness. In this study, we propose polymersomes to enhance the anti-inflammatory properties of prednisolone disodium phosphate (PDP), while reducing the widespread off-target effects and the drug-resistance for the treatment of inflammatory disorders. Particularly, we used pH responsive polymersomes made of amphiphilic poly(2-methacryloyloxyethyl phosphorylcholine)–poly(2-(diisopropylamino)ethyl methacrylate) (PMPC–PDPA) diblock copolymers. The biocompatible nature of the PMPC block endows polymersomes with similar stealth properties to poly(ethylene glycol), thus being resistant to unspecific protein adhesion [13,14,15]. Additionally, PMPC binds to the scavenger receptor B type I (SR-BI) highly expressed in macrophages, which in turn enables the selective binding and internalisation of PMPC–PDPA polymersomes as we recently observed [13,14,15,16]. Whereas, the PDPA block bestows the pH responsive nature of polymersomes, as at pH lower than 6.2 (the PDPA pK_a_) enables the polymersomes disassemble and hence release of loaded drug [17]. We have amply demonstrated [14,17,18,19,20,21,22] that such feature enables the intracellular cargo delivery, where the polymersome disassembly is confined within endosomes. Hence, the associated increase in osmotic pressure allows its endosomal escape and the release of loaded PDP molecules within the cell cytosol. In addition to this, during inflammation, particularly the most severe, the local environment pH can drop as low as to six due the cellular acidosis [23]. Herein, we studied the in vitro cellular uptake of PMPC–PDPA polymersomes, loaded PDP targeted release and hence its therapeutic effect in human macrophages.

## 2. Materials and Methods

### 2.1. Preparation of PMPC–PDPA Polymersomes

The PMPC_25_–PDPA_68_ and Cyanine5–PMPC_25_–PDPA_70_ copolymers were synthesised by atom-transfer radical polymerisation (ATRP) according to a previously published protocol [24,25] (details on supplementary materials about the experimental procedures and characterization).

Polymersomes made by self-assembly of the PMPC_25_–PDPA_68_ copolymers (Appendix A) in aqueous solution were prepared under sterile conditions using the film rehydration method as previously reported [13,21,26,27,28] with some modifications. Briefly, 25 mg of PMPC_25_–PDPA_68_ copolymer was dissolved in an organic solution of 2:1 (*v/v*) chloroform:methanol (Sigma-Aldrich, Dorset, UK). The organic solvents were evaporated under vacuum oven at 38 °C for 24 h, resulting in a uniform polymeric thin film. Then, the film was rehydrated with Dulbecco’s phosphate-buffered saline (DPBS, Sigma-Aldrich, Dorset, UK) at stable neutral pH for a final concentration of 5 mg/mL, and left under continuous magnetic stirring (200 rpm, RT15 power, IKA-Werke GmbH & Co. KG, Staufen, Germany) at room temperature for eight weeks.

Formulations of PMPC–PDPA polymersomes loading prednisolone disodium 21-phosphate (PDP, C_21_H_27_Na_2_O_8_P, MW 484.39, λ (250 nm), Tokyo Chemical Industry Co., Ltd., Oxford, UK) were prepared also using the film rehydration method by dissolving the drug in methanol during the organic phase prior to copolymer film formation. Additionally, for imaging fluorescence techniques, Cy5–PMPC–PDPA polymersomes were prepared using above-mentioned film rehydration method. Mainly, during the organic phase 10% (*w/w*) Cy5–PMPC_25_–PDPA_70_ (Appendix A) was dissolved together with the PMPC_25_–PDPA_68_ copolymer. Afterwards, all formulations of PMPC–PDPA polymersomes were purified as previously described [29] firstly by centrifugation (2000 rcf, 20 min, 20 °C), following the size exclusion chromatography.

### 2.2. Characterisation Studies

#### 2.1.1. Size and Size Distribution Study

The hydrodynamic diameter (D_h_) and polydispersity index of PMPC–PDPA polymersomes were determined by dynamic light scattering (DLS) using the Zetasizer Nano ZS (Zen1600, Malvern Instruments Ltd., Worcestershire, UK) with a 633 nm HeNe laser in a scattering angle of 173°. Measurements were recorded at 20 °C in triplicate and samples were diluted for an average count rate between 100–300 kcps.

#### 2.1.2. Morphology Study

The morphology and size of PMPC–PDPA polymersomes were analysed using the JEOL 100CX II transmission electronic microscope (TEM) (JEOL, Welwyn Garden City, UK) with a tungsten-filament 100 kV, and equipped with a Gatan Erlangshen ES500W camera. Prior to TEM imaging, samples were stained with 0.75% (*w/v*) phosphotungesteic acid (Sigma-Aldrich, Dorset, UK) in earlier prepared glow-discharged copper grids (Agar Scientific, Essex, UK), following a previously reported protocol [30].

#### 2.1.3. Polymer and Drug Quantification Study

The amount of PMPC_25_–PDPA_68_ copolymer and loaded drug after purification process was determined by high-performance liquid chromatography (HPLC) using a Dionex Ultimate^®^3000 (Thermo Scientific, Dartford, UK) This instrument is equipped with a variable wavelength detector and a C18 analytical column (Jupiter Phenomenex 300A, 150 × 4.6 mm, 5 μm). The samples, previously diluted in DPBS at pH 2.0, were run according to a ramp gradient of eluent A [0.05% (*v/v*) trifluoroacetic acid (TFA, Thermo Fisher, Dartford, UK) in Milli-Q filtered H_2_O] and eluent B [0.05% (*v/v*) TFA in methanol]: For 10 min from 5% to 100%; kept constant for 15 min and returned within 1 min to initial condition. By using the Chomeleon software (version 6.80, Thermo Scientific, Dartford, UK), the peak area was integrated at respective elution time and λ on the HPLC system to analyse the UV absorption of PMPC_25_–PDPA_68_ at λ (220 nm) and PDP at λ (250 nm). The amount of PMPC_25_–PDPA_68_ copolymer and PDP was quantified using their respective calibration curves. Further analysis on the drug encapsulation and loading efficiencies were determined using a previously reported method [18] (details on Appendix A).

#### 2.1.4. Drug Release Study

The PDP release study was performed using the dialysis method under sink conditions as previously described with some modifications [31]. Briefly, 1 mL of PDP loaded PMPC–PDPA polymersomes or free drug (at the same concentration of the loaded one) was filled in a cellulose ester dialysis membrane tube (3.5–5 kDa MWCO, Float-a-Lyzer G2, Spectrum Laboratories Inc., Sigma-Aldrich, Dorset, UK). Then, it was dialysed against 10 mL of outer buffer solution under continuous magnetic stirring (200 rpm, RT15 power, IKA-Werke GmbH & Co. KG, Staufen, Germany) at 37 °C for 50 h. These dialyses were carried out against three different outer buffer pH conditions: PBS solution at pH 7.4; acetate-buffered solution at pH 6.5 and 5.0. At regular time points, aliquots (200 μL) were withdrawn and the same volume of respective fresh buffered solution were replaced in order to maintain the sink conditions. The quantification of permeated drug aliquots throughout the 50 h was determined by measuring the UV absorbance of PDP at λ (250 nm) using the UV-Vis microplate spectrophotometer (Synergy HT, Biotek, Swindon, UK) and normalised to the free drug release profile. Mathematical models for drug-release kinetics (Appendix A), including zero-order and first-order equations, Higuchi, Hixson–Crowell and Korsmeyer–Peppas models, were applied to each obtained profile to evaluate the mechanism of drug release (details on Appendix A). The fitting of each model was evaluated based on the correlation coefficient (*r*^2^) values for each model fit.

### 2.3. In Vitro Cellular Studies

#### 2.3.1. Cell Culture and Differentiation

Human leukemic monocytes (THP-1) were cultured and maintained in RPMI-1640, containing 2 mM l-glutamine, 25 mM Hepes (Sigma-Aldrich, Dorset, UK) and supplemented with 10% (*v/v*) heat-inactivated fetal bovine serum (FBS, Sigma-Aldrich, Dorset, UK), 1% (*v/v*) penicillin-streptomycin (Sigma-Aldrich, Dorset, UK), and 0.1% (*v/v*) amphotericin B (Sigma-Aldrich, Dorset, UK). THP-1 cells were used for all in vitro experiments between passage numbers nine and twenty.

THP-1 Blue^TM^ NF-κB reporter cells purchased from InvivoGen were maintained in the same cell culture medium plus supplemented with 100 μg/mL Normocin™ (InvivoGen, Toulouse, France). In addition, 10 μg/mL blastacidin (InvivoGen, Toulouse, France) was added to the growth medium every two passages to maintain selective pressure. The in vitro experiments with these cells were carried out between passage numbers three and nine.

Prior to all in vitro cellular studies, THP-1 cells differentiation into mature macrophages-like state (M0-macrophages) was induced through incubation with 10 ng/mL of phorbol 12-myristate 13-acetate (PMA, Sigma-Aldrich, Dorset, UK) for 48 h in a humidified atmosphere, 95% air, 5% CO_2_ at 37 °C [6].

#### 2.3.2. Cell Viability Assay

For the cell viability assay, THP-1 cells were seeded at a concentration of 5 × 10^3^ cells per well in 96-well plates (CytoOne) and differentiated as mentioned above. Increasing concentrations of PMPC–PDPA polymersomes from 1.3 to 2.0 mg/mL were then incubated for 24 and 48 h in a humidified atmosphere, 95% air, 5% CO_2_ at 37 °C. In another experiment, M0-macrophages were incubated with PMPC–PDPA polymersomes (<600 µg/mL) loaded with PDP in raging concentrations 2.5 to 40 µg/mL for 24 and 48 h in a humidified atmosphere, 95% air, 5% CO_2_ at 37 °C. Additionally, free PDP in PBS solution raging the same concentrations of the loaded one was also incubated with M0-macrophages for 24 and 48 h. Control wells were incubated with equivalent volumes of cell culture medium and/or a solution of 10% (*v/v*) dimethyl sulfoxide (DMSO, Sigma-Aldrich, Dorset, UK) in DPBS. Later on, to evaluate the cytotoxicity of all treatments, the Real-Time-Glo™ MT Cell Viability Assay (Promega Corporation, Hampshire, UK) was used following the manufacturer protocol.

#### 2.3.3. Cell Uptake Imaging

Cell imaging was performed using confocal laser scanning microscopy (CLSM, Leica SP8, Milton Keynes, UK). Firstly, THP-1 cells were seeded at a concentration of 5 × 10^4^ cells per glass-bottom Petri dish (Ibidi) and differentiated as above mentioned. Cy5–PMPC–PDPA polymersomes (0.6 mg/mL) were then incubated for 0.5, 1, 2, 4, 6, 12, 24, and 48 h, in a humidified atmosphere, 95% air, 5% CO_2_ at 37 °C. After each incubation time point, followed by three steps of DPBS washing, M0-macrophages were stained for CLSM live imaging. Hoechst 33342 (Sigma-Aldrich, Dorset, UK) and far-red Cell Mask™ (Life Technologies Ltd., Thermo Fisher Scientific, Renfrew, UK) were used for nuclear and cell membrane staining, respectively. At least 10 different regions of the Petri dishes were captured and analysed using the Fiji ImageJ software (version 2.0). For the quantification of Cy5–PMPC–PDPA polymersomes within macrophages, their fluorescent intensity signal was normalised relative to the nuclear intensity signal.

#### 2.3.4. NF-κB Signalling Imaging and Quantification Assay

NF-κB signalling imaging was preformed using CLSM. Firstly, THP-1 cells were seeded at a concentration of 5 × 10^4^ cells per glass-bottom Petri dish (Ibidi) and differentiated as above mentioned. Followed by M0-macrophage activation to M1 state, which was induced with incubation of 600 ng/mL of lipopolysaccharide (LPS, Sigma-Aldrich, Dorset, UK) for at least 6 h in a humidified atmosphere, 95% air, 5% CO_2_ at 37 °C [6]. Then, M1-macrophages were incubated with PMPC–PDPA polymersomes (0.6 mg/mL), either free PDP or PDP loaded polymersomes (10 µg/mL) for 24 h in a humidified atmosphere, 95% air, 5% CO_2_ at 37 °C. Following treatment, cells were washed with DPBS and fixed using 3.7% formaldehyde (Sigma-Aldrich, Dorset, UK) for 10 min at room temperature (RT). After fixation step, followed by DPBS washing for the membrane permeabilization step, cells were incubated with 0.2% Triton-X (Sigma-Aldrich, Dorset, UK) for a further 10 min at RT. Then, the immunostaining blocking was performed using 5% bovine serum albumin (BSA) (Sigma-Aldrich, Dorset, UK), to prevent unspecific antibody binding. After 1 h at RT, cells were incubated with NFκB p65 Antibody (F-6) Alexa Fluor^®^ 647 (Santa Cruz Biotechnology Inc., Heidelberg, Germany) diluted in 1% BSA overnight in a humidified chamber at 4 °C. The following day, cells were washed with DPBS and the nucleus was stained with Hoescht 33342 (Sigma-Aldrich, Dorset, UK) for 10 min at RT, before visualisation under CLSM. At least 10 different regions of the petri dishes were acquired and the NF-κB nuclear translocation imaging analysis was evaluated by co-localisation (Pierce’s coefficient values) of the NF-κB and nucleus fluorescence intensity signals using Fiji ImageJ software (version 2.0). Additionally, the quantification of macrophages inflammation levels related to the NF-κB signalling activity was carried out in the THP-1 Blue™ NF-κB reporter cells, as previously reported [32]. These cells are stably transfected and express a secreted embryonic alkaline phosphatase (SEAP) reporter gene driven by an interferon-β minimal promoter fused to five copies of the NF-κB transcription factor, which promotes cytokines production. Then, once the NF-κB translocation from the cytosol to the nucleus induces the secretion of the SEAP, this can be quantified by the absorbance read at λ (570 nm) using the UV-Vis microplate spectrophotometer (ELx800, BioTek, Swindon, UK). For the SEAP assay, THP-1 NF-κB reported cells were seeded at a concentration of 5 × 10^3^ cells per well in 96-well plates (CytoOne) and differentiated as above-mentioned. Followed by M1-macrophage activation with 600 ng/mL of LPS for 6 h [6], cells were then incubated with each one of the treatments for more than 6 and 24 h in a humidified atmosphere, 95% air, 5% CO_2_ at 37 °C. The detection and quantification of SEAP activity from the collected supernatant was then preformed according to the manufacturer.

#### 2.3.5. RNA Extraction, Reverse Transcription, and Real-Time Quantitative Polymerase Chain Reaction

Analyses on the gene expression of inflammation-related markers, including TNFα, IL1β, IL6 and IL8, was assessed using real-time quantitative polymerase chain reaction (RT-qPCR). Firstly, THP-1 cells were seeded at a concentration of 10^6^ cells per well in 96-well plates (CytoOne) and differentiated as above-mentioned. Following the M1-macrophage activation with 600 ng/mL of LPS (non-treated control), cells were incubated either with PMPC–PDPA polymersomes (0.6 mg/mL), free PDP or PDP loaded polymersomes (10 µg/mL) for 6 h in a humidified atmosphere, 95% air, 5% CO_2_ at 37 °C. Cells were then lysed and the RNA was extracted following the RNeasy mini kit (Qiagen, Manchester, UK) protocol pre-installed in the QIAcube (Qiagen, Manchester, UK). The total RNA concentration was measured with NanoDrop 8000 spectrophotometer (Fisher Scientific, Dartford, UK). Complementary DNA (cDNA) was synthesised from every 1 μg of total mRNA in 20 μL volume with QuantiTect Reverse Transcription Kit (Qiagen, Manchester, UK) according to the manufacture’s protocol. This procedure provided a fast and efficient cDNA synthesis with integrated removal of genomic DNA contamination. Briefly, the sample of RNA is incubated at 42 °C for 2 min to effectively remove containing genomic DNA, then the reaction occurred for another 15 min at 42 °C and then inactivated at 95 °C. RT q-PCR reaction was performed on yield cDNA synthesised from each sample using QuantiTec^®^ Rotor-Gene™ SYBR Green RT-PCR kit (Qiagen, Manchester, UK) using the Qiagility instrument software (Qiagen, Manchester, UK). This software enables rapid and high-precision system of sample preparation for RT-qPCR analysis, providing a step-by-step guidance for automatic calculation of all primers, cDNA template and Rotor-Gene SYBR Green master mixes needed for the reaction. The list of designed primers of each target gene and reference gene used for the gene expression experiments is in the Appendix A. Following sample preparation, the PCR mixtures are placed in the Rotor-Gene Q cycler (Qiagen, Manchester, UK) and amplification process starts using the following protocol steps: Initial cycling step at 95 °C during 5 min for the DNA polymerase activation; followed by 40 cycles of 95 °C during 5 s for denaturation; and 60 °C during 10 s for combined annealing and extension for all primers. RT-qPCR data analysis of folds-changes in gene expression levels was determined by the—ΔΔCt method (details in supplementary materials), using cycle threshold (Ct) values acquired from the amplification curve using the Rotor-Gene Q instrumentation software (Qiagen, Manchester, UK).

#### 2.3.6. Enzyme Linked Immunosorbent Assay

IL6 and TNFα protein levels secreted by macrophages were quantified by the enzyme linked immunosorbent assay (ELISA). Firstly, THP-1 cells were seeded at a concentration of 10^6^ cells per well in 96-well plates (CytoOne) and differentiated as above mentioned. Following the M1-macrophage activation with 600 ng/mL of LPS (non-treated control), cells were incubated either with PMPC–PDPA polymersomes (0.6 mg/mL), free PDP or PDP loaded polymersomes (10 µg/mL) for 24 h in a humidified atmosphere, 95% air, 5% CO_2_ at 37 °C. Cell supernatants were collected and then the ELISA technique was used following the manufacturer protocol (Invitrogen, Thermo Scientific, Dartford, UK).

### 2.4. Statistical Analysis

Statistical analyses were performed using GraphPad Prism (version 8.2.1). The difference between three and more groups was, respectively, analysed through one-way or two-way ANOVA multiple comparisons. Differences between two groups was evaluated by two-tailed Student’s *t*-test. The differences were statistically significant when * *p* < 0.05, ** *p* < 0.01, *** *p* < 0.001 and **** *p* < 0.0001.

## 3. Results and Discussion

### 3.1. PMPC–PDPA Polymersomes Are Suitable Nanocarriers for PDP

The size of PMPC–PDPA polymersomes is one of the key factors to predict both their stability and behaviour in vivo, including circulation time, biodistribution, cell binding affinity, and uptake [20,22,33,34,35]. All the formulations of polymersomes were characterised in terms of size, distribution, and morphology by DLS and TEM. Light scattering analyses revealed that the loading of PDP drug into PMPC–PDPA polymersomes corresponds to an increase in size from an average hydrodynamic diameter (D_h_) of 117 ± 5 to 178 ± 4 nm (Appendix A). Such a change can be attributed in part to the large number of drug molecules loaded within the polymersomes (defined as the drug loading efficiency; Appendix A) and consequent swelling of the vesicles membrane. In addition, the amount of encapsulated PDP within PMPC–PDPA polymersomes, quantified using HPLC, resulted in 12% ± 4% of drug encapsulation efficiency (details in Appendix A). TEM analyses of PMPC–PDPA polymersomes confirmed the formation of spherical vesicles with D_h_ size in agreement with the DLS measurements (Figure 1a, top image). While, TEM imaging of the PDP loaded polymersomes revealed the formation of vesicles with an oblate shape (Figure 1a, bottom image). The non-spherical nature of the polymersomes hinders the interpretation of the DLS size distribution and while the size increase can be still attributed to the drug loading, the tubular shapes might not be fitted by the traditional models based on spherical geometry. The PDP structure (Appendix A) is partly polar and partly apolar (cholesterol-like) and such amphiphilic nature drives its insertion within the PDPA hydrophobic membrane of polymersomes. Additionally, the PDP anionic phosphate group could potentially interact with some residual positive charges still present in the PDPA block. Both interactions favour the encapsulation process and thus justify the very high loading numbers of drug molecules per polymersome (Appendix A). In a previous work [26], we demonstrated that the inclusion of cholesterol, a molecule similar to PDP, within PMPC–PDPA polymersomes during the film hydration method stabilises the formation of tubular vesicles. Thus, we infer that the PDP insertion results in the formation of oblate polymersomes.

### 3.2. PMPC–PDPA Polymersomes Enable a pH-Controlled Drug Release

To prove the ability of PMPC–PDPA polymersome to release the PDP as a function of the environment pH, we measured the drug release under different pH conditions at 37 °C, mimicking the physiologic pH 7.4, the inflamed extracellular tissue (pH 6.5), and the intracellular compartments (pH 5.0). For PDP loaded polymersomes dialysed under conditions at pH 5.0, resulted in a biphasic drug release profile (Figure 1b) with an initial burst (immediate pulse of 55% within 1 h) followed by a sustained drug release overtime. As expected, under pH 5.0 conditions the PDPA amino groups (pKa 6.2; Appendix A) get protonated and positively charged. The change of polarity makes the PDPA hydrophilic and hence the PMPC–PDPA copolymers are no longer capable of forming polymersomes driving a fast disassembly process [14,17,18,19,20,21,22] hence, resulting in a burst release of the majority of the PDP. The following sustained release suggests a mild interaction between the now cationic PDPA and the anionic PDP phosphate group. Results at pH 6.5 (Figure 1b) also suggest a timed sustained drug release profile (up to 22%) starting after 8 h under dialysis. The two effects are highly desirable within both the macrophage intracellular or inflamed milieu where the drug needs to be freed. The drug release profiles were analysed using several mathematical models (Appendix A): Zero-order, first-order, Hixson–Crowell, Higuch, and Korsmeyer–Peppas. The regression coefficient (*r*^2^) analyses (Appendix A) confirm the Higuchi and zero-order as the best fitting model for pH 5.0 and pH 6.5, suggesting that the PDP release from the PMPC–PDPA polymersomes is indeed caused by a controlled diffusion mechanism [36,37]. This study proves not only the acidic pH responsiveness of the PDPA block to bestow the rapid disassemble of polymersomes, but also the drug release profile (Figure 1b) demonstrated the high stability of PMPC–PDPA polymersomes at physiologic pH 7.4.

### 3.3. PMPC–PDPA Polymersomes Enable a Rapid Intracellular Drug Delivery

PDP, similar to most glucocorticoids, downregulate inflammatory genes reversing histone acetylation binding to glucocorticoid receptors (GR) and thus triggering the recruitment of histonedeacetylase-2 (HDAC2) to the transcription complex [8]. Thus, the site of action of PDP is indeed the cytosol. Hence, the dynamics of cellular uptake of polymersomes by macrophages is a key parameter to achieve efficient intracellular delivery [35]. Cellular uptake studies of Cy5–PMPC–PDPA polymersomes (Cy5–Psome; Appendix A) was assessed using live imaging CLSM. The amount of polymersomes increases in a time-dependent manner up to 48 h, resulting in a bimodal profile (Figure 1c), with a rapid internalisation within 3 h followed by a plateau. The PMPC–PDPA polymersomes bind very quickly to the cell surface, saturating their receptors and dwell at the cell membrane for the first 30 min, as confirmed by the co-localisation of the merged fluorescence signals (Figure 2). A rapid and enhanced cellular uptake of PMPC–PDPA polymersomes was previously reported [15,20,21]. The internalisation mechanism is possibly due the combination of the intrinsic professional phagocytic activity of macrophages [3,4,5] and with the high affinity interaction of the PMPC block towards the scavenger receptor class B type I (SR-BI) [13,14,15,16]. This enables polymersomes to selectively bind to the SR-BI cell receptors, a process that triggers rapid internalisation within macrophages via endocytosis [13,14,15,19,20,21,22].

An important aspect to take into consideration is the potential effect of polymersomes towards cell viability, because of such an enhanced cell uptake level. Most importantly such a fast entry does not affect cell viability in M0-macrophages. Both 24 and 48 h of incubation time with PMPC–PDPA polymersomes showed (Figure 1d; Appendix A) no significant effects on cells survival, and no cytotoxicity was observed up to a concentration of 1 mg/mL. Further cytotoxicity investigations were carried out aiming to compare M0-macrophages viability as a function of treatment with either the PDP loaded PMPC–PDPA polymersomes or the free drug (both tested using the same PDP final concentration), after 24 and 48 h of incubation. M0-macrophages treated with PDP loaded polymersomes exhibited over 80% of cell viability upon 24 h of incubation time (Figure 1e). The cytotoxic effect of the free PDP was comparable to the loaded one only at lower concentrations (<10 µg/mL). In contrast, cells viability was reduced in a concentration-dependent manner after 48 h of incubation for both free and loaded PDP (Appendix A), resulting in 30% decrease of cell viability at the highest concentration.

### 3.4. PDP Loaded PMPC–PDPA Polymersomes Promote Inflammation Resolution In Vitro

In vitro cellular studies were carried out to evaluate the anti-inflammatory effect of PDP loaded PMPC–PDPA polymersomes in activated M0-macrophages (also known as M1-macrophages). We stimulated M0- into M1-macrophages by exposing them to LPS and their activation was assessed by checking for the nuclear translocation of the pro-inflammatory transcription factor NF-κB [7,38]. As shown by immunofluorescence microscopy (Figure 3a), the NF-κB transcription factor is mainly cytosolic within M0-macrophages and upon LPS stimulation the NF-κB translocate to the nucleus giving rise to M1-macrophages. Such an assay can be easily quantified by imaging and we measured 80% cells presenting nuclear NF-κB fluorescence intensity in good agreement with previously reported data [38]. The NF-κB activity was also measured using THP-1 genetically modified with a NF-κB reporter that upon translocation secretes in the media alkaline phosphatase [38]. The secreted embryonic alkaline phosphatase (SEAP) assay allows the detection and quantification of NF-κB activation through the quantification of the phosphatase level by colorimetry. Both assays allow for assessing anti-inflammatory activity of free PDP, loaded, and unloaded PMPC–PDPA polymersomes. Quantitative imaging analysis of immunofluorescence macrographs (Figure 3b) revealed a mild anti-inflammatory effect of unloaded polymersomes and a reduction to 60% with the free PDP. While PDP loaded polymersomes significantly reduce (*p* < 0.0001 compared to control) the nuclear translocation of NF-κB by 50% after 24 h incubation. Results plotted in Figure 3c showed that all treatments reduced the amount of SEAP secreted with respect to the M1-macrophages (used as control) either after 6 or 24 h of incubation. Remarkably, both unloaded and PDP loaded PMPC–PDPA polymersomes show a significant reduction on SEAP production after 6 h of incubation. This reduction was much more significant (*p* < 0.001 compared to control) after 24 h of treatment of M1-macrophages with PDP loaded polymersomes. We thus demonstrated that PMPC–PDPA polymersomes mediated intracellular delivery of PDP and enhances its anti-inflammatory effect. The inhibition on the NF-κB transduction signalling pathway is indeed significantly higher (*p* < 0.05 compared to free PDP). In the early stages of the inflammation, Nf-κB translocation from the cytosol to the nucleus occurs, as confirmed by the differences in the CLSM imaging analysis between M0- and M1-macrophages (Figure 3a). The translocation of Nf-κB induces then the upregulation of pro-inflammatory genes, TNFα, IL1β, IL6, and IL8 [7,10,38]. We thus measured such genes mRNA expression by RT-qPCR (Figure 3d) confirming that the transition from M0 to M1 corresponds with upregulation of all these pro-inflammatory genes [6,11]. Further analyses of the RT-qPCR data plotted in Figure 3d revealed that 6 h incubation with any of the treatments significantly decreases gene expression levels of all inflammation-related markers compared to the control. Most notably, all the treatments equally decrease gene expression levels of IL8 (a potent neutrophil chemo-attractant [39]) by 2-fold (*p* < 0.0001 compared to control). Such data suggest that PDP loaded PMPC–PDPA polymersomes can also potentially regulate the activation of neutrophils in vivo, and hence modulate the recruitment of leucocytes to the inflammation site. Both RT-qPCR and ELISA results confirmed the well-known immunosuppressive and anti-inflammatory effect of PDP [8,12]. The free PDP indeed significantly (*p* < 0.0001 compared to control) decreases the gene expression of all the tested pro-inflammatory cytokines (Figure 3d). Likewise, PDP loaded PMPC–PDPA polymersomes significantly downregulate the IL1β and IL6 gene expression levels (more than 10-fold for *p* < 0.0001 compared to control). Particularly, IL6 plays not only an important role in the activation of both innate and adaptive immune system, but it is also involved in the regulation of chronic inflammatory response [10,40]. ELISA analysis plotted in Figure 3e further revealed that IL6 protein secretion levels are significantly decreased (*p* < 0.0001 compared to control) after 24 h of incubation with PDP, either loaded or as free drug. Similar behaviour was observed for TNFα secretion levels (Figure 3e), which significantly decreases (*p* < 0.001 compared to control) upon incubation with any of the treatments. Though, and in agreement with the gene expression studies, ELISA results also show that there was not a significant difference between free and loaded PDP within PMPC–PDPA polymersomes. In conclusion, PDP loaded PMPC–PDPA polymersomes have a significant effect on the inhibition of the NF-κB nuclear translocation (Figure 3c). They also modulate the gene expression profiles of pro-inflammatory mediators (Figure 3d), and the secretion level of IL6 and TNFα proteins (Figure 3e) by M1-macrophages.

Remarkably, all the in vitro inflammation-related studies (Figure 3) suggest that unloaded PMPC–PDPA polymersomes inactivated the NF-κB signalling pathway in M1-macrophages and, as consequence, the gene expression of inflammation-related cytokines. We believe that this might be related to the phosphatidylcholine anti-inflammatory properties [41] possibly associated with its ability to target the SRBI cell receptor. It was previously reported [42] that this receptor regulates macrophage inflammation upon ligand binding, as the activation of the NF-κB is reduced and leads to the increase of anti-inflammatory mediators. Still, future studies need to be performed to understand these remarkable evidences. Taken together, with these in vitro studies, we confirm that LPS-activated NF-κB mediates the inflammatory response in macrophages. Moreover, we demonstrate that PDP loaded PMPC–PDPA polymersomes inhibit the NF-κB nuclear translocation pathway and hence reduce inflammation-related expression at molecular level.

## 4. Conclusions

PMPC–PDPA polymersomes have been demonstrated here to be a suitable drug delivery carrier enabling both drug’s protection and stability overtime under mimicking physiologic pH conditions. The drug release study confirmed the pH responsiveness of polymersomes to acidic pH, mimicking the intracellular endosomal compartments. This enables an efficient intracellular and targeted delivery of loaded prednisolone into the cytosol. In vitro cell uptake studies revealed that PMPC–PDPA polymersomes enhance the accumulation of prednisolone within macrophages without affecting their viability. Plus, this increases the intracellular bioavailability of effective low dosage of PDP and hence its anti-inflammatory effect. Indeed, we show that the inflammation-activated NF-κB signalling pathway was shut down in inflamed macrophages treated with PDP loaded within polymersomes, and as consequence the expression of pro-inflammatory genes and proteins is reduced. Overall, pH-responsive PMPC–PDPA polymersomes ensure stability and selectivity towards target cells, as well as promote inflammation resolution in vitro. We prove the therapeutic potential of pH-responsive polymersomes to reduce the well-known deleterious side effects and the resistance that compromises the effectiveness of glucocorticoid therapy in the treatment of inflammatory disorders.

## Figures and Tables

**Figure 1 pharmaceutics-11-00614-f001:**
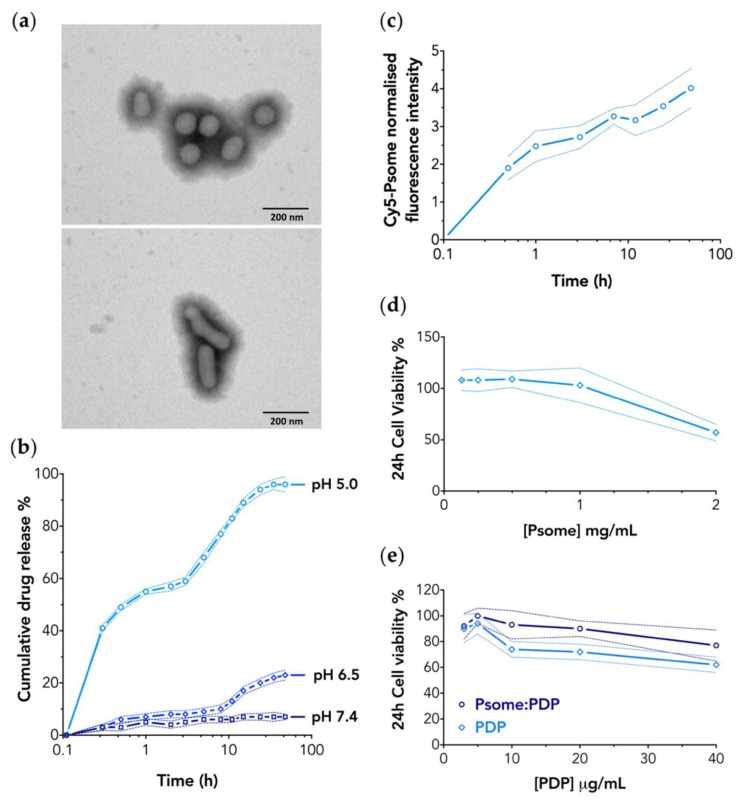
(**a**) Representative TEM images of empty (top) and PDP loaded PMPC–PDPA polymersomes (bottom) produced via film rehydration method (200 nm scale bar). (**b**) PDP cumulative drug release profile in each tested pH condition over 50 h at 37 °C. Data express as mean ± SD (*n* = 3). (**c**) Cy5–PMPC–PDPA polymersomes normalised fluorescent intensity relative to the nucleus signal measured as a function of time upon uptake by macrophages. Data express the mean ± SD (10 images for *n* = 2). Cell viability assay after 24 h incubation with increasing concentrations of (**d**) unloaded PMPC–PDPA polymersomes (Psome), (**e**) either free PDP or PDP loaded polymersomes (Psome:PDP). Data express as mean ± SD (*n* = 3).

**Figure 2 pharmaceutics-11-00614-f002:**
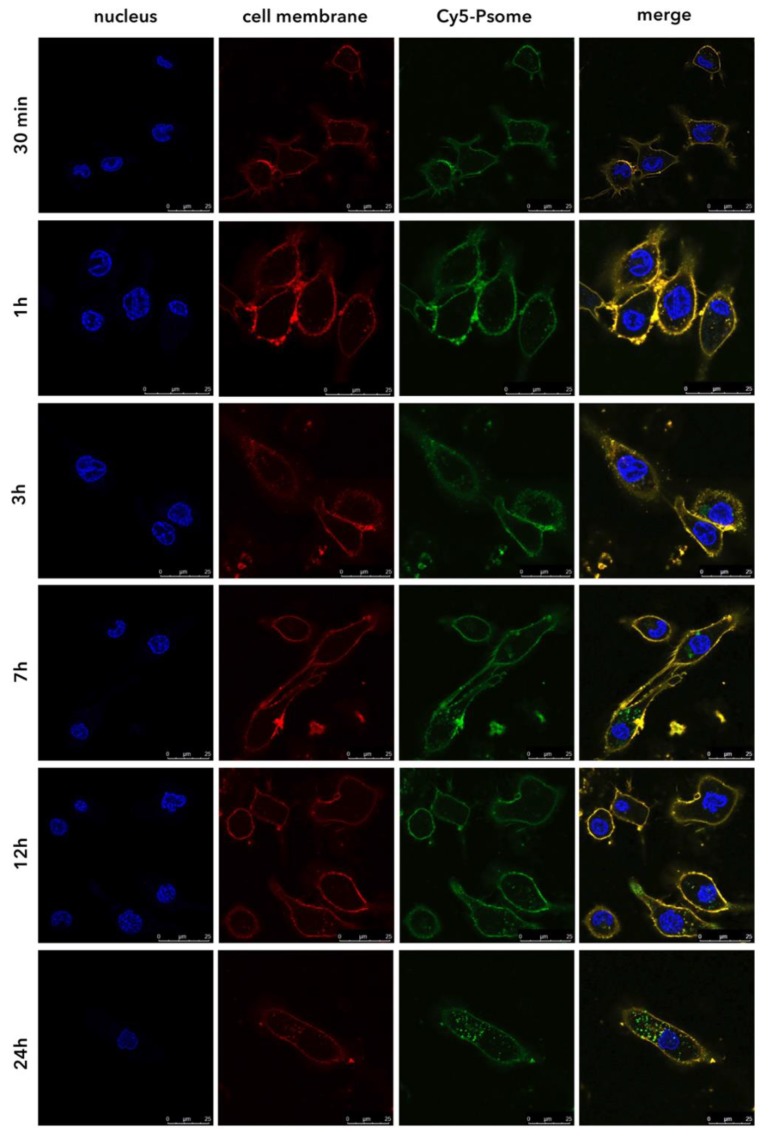
Representative CLSM images of Cy5–PMPC–PDPA polymersomes (green fluorescence intensity signal) cellular uptake overtime (scale bar: 25 μm). Staining of the cell nuclei (blue fluorescence intensity signal) with Hoechst 33342 and cell membrane (red fluorescence intensity signal) with far-red CellMask™. The yellow fluorescence intensity signal corresponds to the co-localisation (merge) of the Cy5–PMPC–PDPA polymersomes and cell membrane fluorescence signals.

**Figure 3 pharmaceutics-11-00614-f003:**
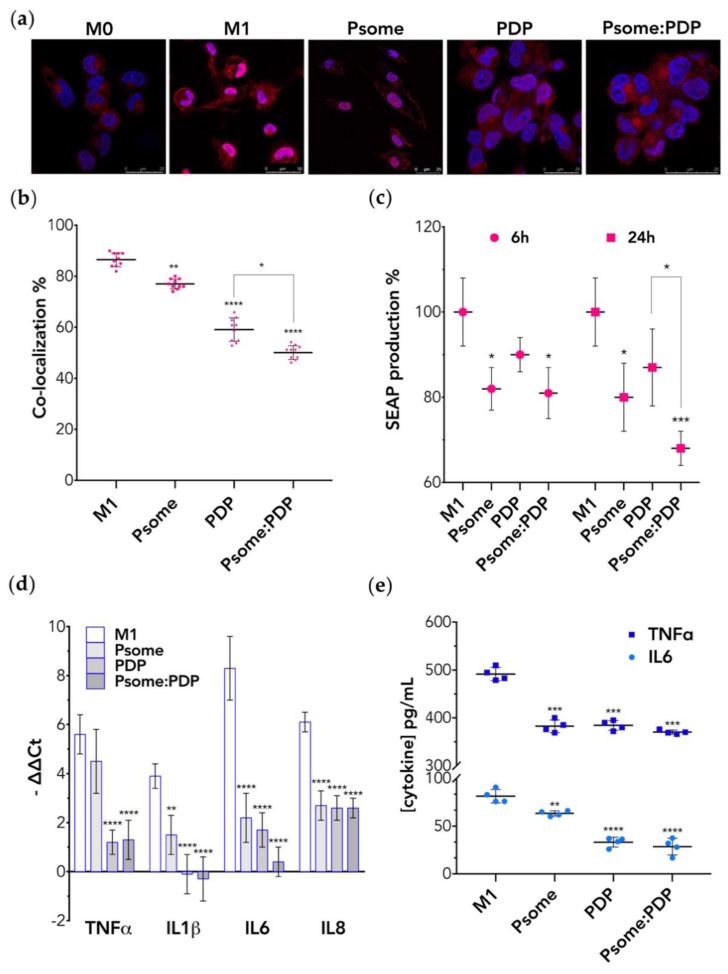
In vitro inflammation-related cellular studies on M1-macrophages after incubation with either PMPC–PDPA polymersomes (Psome), free PDP, or PDP loaded polymersomes (Psome:PDP). (**a**) Representative CLSM images of NFκB (red fluorescence intensity signal) translocation from cytoplasm to the nucleus (blue fluorescence intensity signal) (scale bar: 25 μm). (**b**) Co-localisation analysis of the merged fluorescence signals (pink). Data express as mean ± SD (5 images for *n* = 2). (**c**) SEAP assay for the quantification of Nf-κB nuclear translocation. Data express as mean ± SD (*n* = 3). (**d**) RT-qPCR of pro-inflammatory genes expression levels. Data express as mean ± SD (*n* = 3). (**e**) ELISA for IL6 and TNFα protein secretion levels. Data express as mean ± SD (*n* = 4). In all experiments, the differences were statistically significant when * *p* < 0.05, ** *p* < 0.01, *** *p* < 0.001, and **** *p* < 0.0001.

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
