# Peer review of "Macrophage Targeting pH Responsive Polymersomes for Glucocorticoid Therapy"

_pharmaceutics, 2019, doi:10.3390/pharmaceutics11110614_

Round 1
Reviewer 1 Report
There are a number of imprecise descriptions, above all in the introduction.
Please also check the British/American spelling, e.g. internalisation, localisation or ization.
- introduction
Line 36 “Inflammatory diseases comprise a vast range of disorders and conditions branded by an uncontrolled over-activation of the immune system and henceforth inflammation” As it is written, this is an almost self-justifying statement. I would suggest to remove the last three words (and maybe replace the very strong ‘branded’ with a softer ‘characterized’)
Line 40 “are key players cells” either key players or key cells.
Line 42 “are not only responsible in the patrol for pathogens, but also for the immunomodulation of the inflammatory process” are not only responsible of patrolling pathogens, but also of modulating the inflammatory process (immunomodulation of the inflammation is almost a tautology)
Line 43 “Under inflammatory stimuli, macrophages become activated through the interaction of the Toll-like receptors (class of proteins that play a key role in the innate immune system) with specific cytokines, such as interleukin-1β (IL1β), interleukin-6 (IL6) and tumour necrosis factor-α (TNFα), and/or bacterial lipopolysaccharide.” This is a rather simplistic point of view. First, lipopolysaccharides (plural, it is not a single molecule) are only a specific example of ligands for Toll-like receptors. Second, these cytokines have their own receptors, which when stimulated can lead to macrophage activation also in the absence of Toll-like receptor binding events.
Line 47 “nuclear factor kappa-B (NF-κB) translocation signalling pathway, which is a common pathogenic mechanism in several inflammatory diseases” Pathological, not pathogenic.
Line 50 “These molecules can act as both pro- or anti-inflammatory mediators, and can thus induce either deleterious effects or promote the resolution of inflammation” This sentence conveys a wrong message, i.e. that pro-inflammatory signaling is per se deleterious. On the contrary, they are inherently defensive and can become deleterious.
Line 60 “Despite the effective anti-inflammatory mechanism of action, prednisolone has a very low water solubility…” Indeed it is an effective anti-inflammatory mechanism BECAUSE of its low water solubility, that allows for high affinity for the intracellular steroid receptor. See also my comment re line 269; here a paragraph should be added, because the authors mention the problem of prednisolone poor water solubility, but then use its phosphate derivative, whose water solubility I believe is not so bad.
Line 72. “polyethylene glycol” poly(ethylene glycol).
- Experimental part
Line 150. “For the cell viability assay, THP-1 cells were seeded at a concentration of 5⋅ 10^3 cells/well in a 96-well plate (CytoOne) and differentiated as mentioned above.” Unfortunately, there is above there is no mention of a differentiation protocol (this is indeed where it should be described). Further, these macrophages are described as MPhi-macrophages, but I suspect that the authors meant M0, since MPhi is an abbreviation for macrophages themselves. Please correct as M0 throughout the manuscript.
Line 161. “THP-1 cells were seeded at a concentration of 5⋅ 10^4 cells/well in glass-bottom petri dishes (ibidi) and differentiated above mentioned” Petri and Ibidi have capital letters, ‘differentiated as mentioned above’ (but again, it is not mentioned). Last, the cell density is not clear: I understand that there 50,000 cells per dish (not per well), but the surface of a Petri dish is much more than ten times larger than that of a well of a 96-well plate. Therefore, how lower is the cell density of this experiment?
- Results and discussion
Line 247. “sizes bellow 200 nm” below
Line 252. 117.3 ± 4.5 nm and 177.5 ± 3.6 nm. Sizes should be rounded to the closest 5 nm, due to the poor precision of DLS.
Line 255. “with a uniform Dh size distribution within the range of 100 nm.” Unclear what uniformity ‘within the range of 100 nm’ means.
Line 260. “between the PDP positive charged and PMPC copolymer block”. First, language: the sentence should be ‘between the positively charged PDP and the PMPC block copolymer’ or ‘between the positively charged PDP and the PMPC block in the copolymer’. In either case, A) PDP is negatively (not positively) charged. B) PMPC is zwitterionic and it is kind of difficult to see it its ammonium ions interacting with other phosphates if they have a phosphate just two carbon atoms away…
Line 269-273 and supp info. The drug loading efficiency in this case makes very little sense until one understands that the authors did not use prednisolone, but prednisolone phosphate, which is water-soluble and cannot cross the vesicle membrane. The authors should make this clear in the introduction, where on the contrary simply prednisolone is mentioned.
Figure 1 d and e. In the caption, the authors should recall the PDP/polymer weight ratio.
Line 307. “by the co-localisation merge fluorescence signals” by the co-localisation of the merged fluorescence emissions.
Figure 3d. Between square brackets one should read the name of the species, which n this case should be [cytokine], while the units should be reported after, in round brackets.
Author Response
"Please see the attachment.

Reviewer 2 Report
There was a severe mistake of the title. The term “polymersome” specifically refers to the nanoparticle (NP) with a hollow core made by self-assembly polymers in water with a morphology of vesicle (double layer membrane like the cells). Unfortunately, the TEM images including the PMPC-PDPA and PMPC/PDPA/PDP couldn’t indicate the self-assembled structure of polymersome. The morphology of NPs plays a key role in their cell uptake level. I suggest the authors should supply more reliable proofs (for example AFM or DLS/SLS) to confirm the structure of the NPs. Actually, the NPs seem like the nano-sized spheres commonly seen in literatures according to the TEM images. There is no clear and brief relation of the “Introduction section” with the “titile” of this article. As claimed by the authors, the highlight this article is the “pH-responsive polymersome” for macrophage targeting with GS therapy. Hence, the “Introduction section” should mainly review the progress of the GS delivery to the macrophage with the NPs and the key role of the pH-responsiveness in this field. Unnecessarily, the authors used so many words to describe the general knowledges about the role of macrophage. Finally, the authors should explain the basic mechanism of the pH-triggered PDP release from the PMPC/PDPA NPs by using one or two sentences in the Introduction part. I noticed that the authors had synthesized the PMPC-PDPA and Cy5-PMPC-PDPA with the same chain length by the RAFT and the ATRP, respectively. However, the authors didn’t provide the molecular weight (Mw) of each block (MPMPC/MPDPA) and polydispersity (PDI) characterization with GPC (or SEC) or other tools. Thus, it will be very difficult to let the readers be sure that the PMPC-PDPA and Cy5-PMPC-PDPA have the same self-assembly behaviors in size and morphology regulated by the MPMPC/MPDPA and PDI. As we all know, the size and morphology of NPs can affect the internalization route and then the cell uptake level seriously. Hence, the Cys-5 labeling will fail afterwards to evaluate the uptake level of macrophage.Finally, there are a lot of other problems in this article, which will hurt the quality of this article. For example,
Line 260-261: the statement explaining the cause of enlarged after drug loading was incorrect. PDP is negative charge because of the structure of phosphate. The PDPA block with tertiary amine side chains (pKa~2) will be partially protonated at the pH 7.4. Thus, PDP will be capsulated into the PDP into the hydrophobic core of the PMPC-PDPA NPs by the electrostatic attraction as well as the hydrophobic interaction between the PDP and the PDPA. The increase in size probably results from the huge capsulation of the PDP. The plots in the Fig 1b-1e should be given with error bars, not dot line. The Fig 1d was no necessary and meaningless. Only cell viability with 48 h is enough. The article should be brief. According to the release plots, a brief explanation of the pH-responsiveness of the NPs should be given. In the results and discussion section, the authors prepared the title of each part with a sentence. I think a short phrase will be better. 3 showed that the blank PMPC-PDPA NPs also effectively inhibited the gene expression of each inflammation factors. A deep discussion about this should be provided. Spelling errors and incorrect writing of physical units should be avoided.Author Response
"Please see the attachment.

Reviewer 3 Report
This paper, by Gouveia et. al, studies the ability of pH responsive polymersomes to precisely deliver steroids to the inflamed organ. Authors studied the ability in vitro thus demonstrating the potential of this system as nano carrier for steroids. Many details should be added to the text before this manuscript can be accepted:
Why did the authors use polymersomes (rather than nano particles, liposomes etc.)? Please elaborate. The polymer used to fabricate polymersomes, PMPC25 -PDPA68, should be presented in the text (chemical structure) and more importantly, all related date (NMR, GPC etc.) should be presented. Reference by another group is not enough. Polymersomes were kept stirring for 8 weeks-what was the effect on drug loading? Stability? Why did the author switched from HPLC (used to measure drug loading) to UV-Vis? Please demonstrate that empty polymersomes (w/o drug) didn’t absorbed at 250 nm otherwise repeat this experiment. Please explain the disassemble of the polymersome based on the chemical structure of the polymer. This is the main mechanism and should be better explained.Author Response
"Please see the attachment.

Round 2
Reviewer 2 Report
The authors made a lot modifications about their work. However, the crucial error in the title has not yet been corrected. Eventhough the authors cited a lot of their papers to prove that the self-assembly NPs of the PMPC-PDPA were polymersomes, the TEM pictures (no double layers observed) given by the authors can not support this conclusion. The scientific term must be accurate. So I suggest this article should be rejected at this time, unless the authors supply reliable proofs in this paper to confirm the vesicle-like structure of the NPs.
In addtion, the authors didn't supply the GPC traces to show that the PMPC-PDPA and Cy5-PMPC-PDPA had the nearly equal polymer length. Thus, their precondition that the two polymers may have the same self-assembly behaviors is not believable, which means that the NPs fromed by the two polymers may have different biological properties.
Author Response
"Please see the attachment."

Reviewer 3 Report
I believe the manuscript has been
improved and now warrants publication in Pharmaceutics.
Author Response
thank you very much for the kind assessment
Round 3
Reviewer 2 Report
The authors upload the correct TEM images which clearly shows the polymersome strucure in their response letter. Hence, this paper can be accepted after the minor revision below:
1) This right TEM picture should replace the Figure 1a in the main text and the Figure S2b of the ESI, because both the two pictures can't not distinguish the vesicle structure, and from the two pictures we can also conclude that the NPs are probably nano-spheres.
I understand the authors' impatience about the modification of this paper, but the results in a scientific paper must be supported by the data.
